

# Long-range dependence in coastal framework geology: Asymmetries and implications for barrier island resiliency

Phillipe A. Wernette[1,2,*], Chris Houser[1], Bradley A. Weymer[3], Mark E. Everett[4], Michaelq Bishop[2], Bobby Reece[4]

[1]University of Windsor, Department of Earth and Environmental Sciences, Windsor, Ontario, Canada N9B 3P4
[2]Texas A&M University, Department of Geography, College Station, Texas, USA 77843
[3]GEOMAR Helmholtz Centre for Ocean Research Kiel, 24148 Kiel, Germany
[4]Texas A&M University, Department of Geology and Geophysics, College Station, Texas, USA 77843

*Correspondence to*: Phillipe A. Wernette (wernette@uwindsor.ca)

**Abstract.** Barrier island transgression is influenced by the alongshore variation in beach and dune morphology, which determines the amount of sediment moved landward through washover. While several studies have demonstrated how variations in dune morphology affect island response to storms, the reasons for that variation and the implications for island management remain unclear. This paper builds on previous research by demonstrating that the framework geology can influence beach and dune morphology asymmetrically alongshore. The influence of relict paleo-channels on beach and dune

morphology on Padre Island National Seashore, Texas was quantified by isolating the long-range dependence (LRD) parameter in autoregressive fractionally-integrated moving average (ARFIMA) models. ARFIMA models were fit across all scales and a moving window approach was used to examine how LRD varied with computational scale and location along the island. The resulting LRD matrices were plotted by latitude to place the results in context of previously identified variations in the framework geology. Results indicate that the LRD is not constant alongshore for all surface morphometrics. Many flares in

the LRD plots correlate to relict infilled paleo-channels in the framework geology, indicating that the framework geology has a significant influence on the morphology of PAIS. Barrier island surface morphology LRD is strongest at large paleo-channels and decreases to the north. The spatial patterns in LRD surface morphometrics and framework geology variations demonstrate that the influence of paleo-channels in the framework geology can be asymmetric where the alongshore sediment transport gradient was unidirectional during island development. The asymmetric influence of framework geology on coastal

morphology has long-term implications for coastal management activities because it dictates the long-term behavior of a barrier island. Coastal management projects should first seek to understand how the framework geology influences coastal processes in order to more effectively balance long-term natural variability with short-term societal pressure.

## 1   Introduction

Since modern barrier island morphology is the product of past and present coastal processes acting over pre-existing morphologies, effective barrier island management requires a comprehensive knowledge of how an island has evolved to its current state in order to understand how it may change in the future. Continued sea level rise and future climatic uncertainty





represent significant concerns about the resiliency of barrier islands and threats to many coastal communities (U.S. Environmental Protection Agency, 2016). Barrier island geomorphology can exhibit considerable variability alongshore, leading to varying responses to storm activity that ultimately determines the response of the island to sea level rise. Understanding the source of variability in beach and dune morphology can provide insight into how the barrier island is likely

to change in response to future storms, which ultimately determines how the island will respond to sea level rise.

Storm waves interact with the variable morphology of the nearshore, beach, and dunes to determine patterns of vulnerability along a barrier. To some degree, variations in the nearshore, beach, and dune morphology are influenced by the framework geology (Houser et al., 2008; Houser, 2012; Houser et al., in press; Hapke et al., 2010; Hapke et al., 2016; Riggs et al., 1995).

In this paper, the term "framework geology" is defined as any subsurface variation in geologic structure, where geologic structure can be caused by variations in sediment type (i.e. sand vs. silt), differences in compaction, or significant changes in the subsurface organic content or mineralogy. This term encompasses the subsurface and bathymetric geologic structure (onshore and offshore), which may include rhythmic bar and swale structures (Houser and Mathew, 2011; Houser, 2012), shoreface attached sand ridges (SASR) overlying offshore glacial outwash headlands (Hapke et al., 2010; Schwab et al., 2013),

or buried infilled paleo-channels (Fisk, 1959; Anderson et al., 2016; Simms et al., 2010; McNinch, 2004; Schupp et al., 2006; Browder and McNinch, 2006). Since the framework geology can provide insight into historical patterns of island transgression (Houser, 2012; Houser et al., 2015; Lentz et al., 2013; Hapke et al., 2016), it is vital to better understand how the framework geology influences variability in modern beach and dune morphology. Despite its importance, framework geology remains absent from contemporary barrier island change models that treat the geology as being uniform alongshore (Wilson et al.,

2015; Plant and Stockdon, 2012; Long et al., 2014; Gutierrez et al., 2015). Sections of a barrier island that experience greater washover will experience a net loss of sediment landward and localized erosion, but the dissipative nature of shoreline change (see Lazarus et al., 2011) means that those losses are distributed alongshore. In this respect, the variation in beach and dune morphology alongshore forced by the framework geology can influence the rate of historical shoreline retreat and island transgression and need to be considered in models of barrier island response to sea level rise.

The influence of framework geology on barrier island morphology is well documented by work along the New York, Florida, and North Carolina coasts. Submerged glacial outwash headlands along Fire Island, NY reflected in the nearshore bathymetry as a series of shore-oblique ridges and swales (Hapke et al., 2010; Schwab et al., 2013). The nearshore bathymetry impacts sediment transport gradients along the island, which has implications for beach and dune response and recovery following a

storm. Using sediment cores in conjunction with ground-penetrating radar (GPR) and seismic surveys, Houser (2012) demonstrated that variations in shoreline change patterns, beach width, and dune height corresponded to ridges and swales at Pensacola, FL. Shoreline position was more stable along the ridges, resulting in a wider beach (Houser, 2012). The wider beach, in turn, provided more sediment for onshore winds to create taller and more persistent dunes (Houser, 2012). Paleo-channels dissecting the southeastern U.S. Atlantic coast also align with hotspots of shoreline change (Schupp et al., 2006;



Lazarus et al., 2011). However, Lazarus et al. (2011) argued that "*shoreline change at small spatial scales (less than kilometers) does not represent a peak in the shoreline change signal and that [shoreline] change at larger spatial scales dominates the [shoreline change] signal*". This implies that variations in the framework geology, such as paleo-channels, do not influence long-term shoreline change, but as noted, shoreline change is influenced by the alongshore variation in beach

5 and dune morphology and the dissipative behavior of shoreline change does not negate the importance of framework geology. In addition to framework geology, the alongshore variation in dune morphology is influenced by the distribution of vegetation in both space and time (Lazarus et al., 2011; Lazarus, 2016; Goldstein et al., 2017), creating a self-organized behavior that is ultimately set up by the variation in beach and dune morphology forced by the framework geology (see Houser, 2012; Weymer et al., 2015b; Stallins and Parker, 2003).

The purpose of this paper is to test the hypothesis that relict infilled paleo-channels in the framework geology of a barrier island play a significant role in influencing the alongshore variation in beach and dune morphology at a range of alongshore length scales. Based on the combination of a variable framework geology and a dominant alongshore current it is feasible that the framework geology may influence barrier island geomorphology at discrete spatial scales and that this influence may be

asymmetric. Central to this hypothesis is the idea that the modern island morphology itself is scale-dependent, which has been proposed and supported by previous studies (Lazarus et al., 2011; Lazarus and Armstrong, 2015; Lazarus, 2016; Houser, 2012; Houser et al., 2015). Padre Island National Seashore (PAIS) on North Padre Island, Texas, represents an ideal location to test this hypothesis because previous studies have documented significant variability in the subsurface framework geology (Fig 1; Wernette et al., 2018; Fisk, 1959; Anderson et al., 2016; Weymer, 2016, 2012; Weymer et al., 2018) and there is substantial

alongshore variation in beach and dune morphology. Given that the dominant current along the central Texas coast flowed from north to south during the Holocene (Sionneau et al., 2008), it follows that the dominant alongshore sediment transport gradient during the time also flowed from north to south. It is feasible that paleo-channels along PAIS would have had interacted with the southerly alongshore current and sediment transport to asymmetrically influence barrier island geomorphology during island transgression. In this scenario, areas updrift of a paleo-channel would be distinctly different from

areas downdrift of the paleo-channel, because the channel acts as a uni-directional sediment sink in the coastal sediment budget during island development. Given the complexity of the PAIS framework geology, the results of this paper are valuable to managing coastal resources in other areas with complex underlying and offshore framework geology.

## 2   Methods

### 2.1   Regional Setting

Padre Island National Seashore encompasses a large portion of North Padre Island, the longest continuous barrier island in the world. Located along the south Texas, USA coast, PAIS represents an ideal location to quantify the alongshore influence of framework geology on barrier island geomorphology because of the multiple previously identified paleo-channels dissecting



the island (Fig. 1; Fisk, 1959; Simms et al., 2007; Anderson et al., 2016). Similarly, the modern surface morphology varies alongshore. Central PAIS is characterized by large, relatively continuous dunes, compared to the elongated parabolic dunes along northern PAIS and the heavily scarped and dissected dunes in southern PAIS. Padre Island is separated from the mainland by Laguna Madre, Baffin Bay, and the Intracoastal Waterway (ICW), which was dredged during the 1950s.

Multiple paleo-channels dissect the framework geology of central PAIS and Laguna Madre (Fig. 1; Fisk, 1959). These channels were suggested to have been incised into the Pleistocene paleo-surface and infilled during Holocene transgression. The prevailing theory of formation of PAIS is that the island was initially a series of disconnected barrier islands during the last glacial maximum (~18ka), when a series of channels were incised into the paleo-topographic surface (Weise and White, 1980).
Rapid sea level transgression during the late-Pleistocene and Holocene drown the relict dunes and submerged other dunes located approximately 80 km inland, resulting in disconnected offshore shoals in the current location of PAIS. The disconnected shoals coalesced around 2.8 ka because sand from the relict Pleistocene dunes (~80 km offshore) and sediment discharged from rivers was reworked via alongshore currents, resulting in a continuous subaqueous shoal. Eventually, sediment from offshore relict dunes and increased river discharge supplied enough sediment to the shoals that they aggraded vertically,
becoming subaerially exposed in the same location as the modern barrier island (Weise and White, 1980).

A series of studies in the Gulf of Mexico have focused on extracting a buried ravinement surface, also referred to as the marine isotope stage (MIS) II paleo-surface and buried Pleistocene surface, including the area offshore of PAIS (Fig. 1; Anderson et al., 2016; Fisk, 1959; Simms et al., 2010). Maps of the MIS II surface indicate that PAIS is dissected by at least two substantial
paleo-channels. One large channel dissects PAIS at an oblique angle near "the hole" in Laguna Madre, an area immediately landward of PAIS characterized by consistently deeper water (Fisk, 1959). Based on knick points in the MIS II paleo-surface, this large channel appears to meander from a northeasterly orientation to easterly orientation as it crosses PAIS, eventually flowing into a large paleo-channel adjacent to Baffin Bay. The large paleo-channel forming Baffin Bay is the combined ancestral Los Olmos, San Fernando, and Patronila Creeks (LOSP), which was drowned during sea level transgression and
eventually filled with sediment (Simms et al., 2010). Complexities in the framework geology and modern island geomorphology make PAIS an ideal location to examine how framework geology influences barrier island geomorphology.

Previous studies of PAIS have utilized geophysical surveys and sediment cores to document variation in the depth to a buried Pleistocene paleo-surface (Fisk, 1959;Anderson et al., 2016; Weymer et al., 2016; Wernette et al., 2018). Weymer et al. (2016)
confirmed paleo-channels in the buried Pleistocene paleo-surface using a 100 km alongshore electromagnetic induction (EMI) survey, where areas of lower apparent conductivity are indicative of a deeper buried surface based on the difference in conductivity between overlying Holocene sand and the buried silty clay Pleistocene paleo-surface. Areas where the subsurface apparent conductivity decreased alongshore coincided with paleo-channels which had been previously mapped. Wavelet decomposition of the alongshore EMI survey and offshore bathymetry serve as proxies for the onshore and offshore framework



geology, respectively. When analyzed and interpreted in conjunction with alongshore beach and dune morphometrics, these metrics reveal that larger beach and dune systems are located within the previously mapped paleo-channels (Wernette et al., 2018). The current paper expands on previous research by adapting economic forecast models to determine how paleo-channels in the framework geology have influenced beach and dune evolution and whether this influence is directional and scale-
dependent.

## 2.2    Data sources and validation

Examining the relationships between surface and subsurface barrier island geomorphology requires continuous alongshore data for surface morphology and subsurface framework geology. Barrier island surface morphometrics (*i.e.* beach width, beach
volume, dune toe elevation, dune crest elevation, dune height, dune volume, island width, and island volume) were extracted every 1 m along the entire length of PAIS using an automated multi-scale approach (Wernette et al., 2016). This approach is advantageous because it is less subjective and more efficient than conventional approaches to extracting island morphology. It is important to note that shoreline change was not used in this analysis because previous research demonstrates that the modern beach-dune morphology at PAIS is decoupled from shoreline change processes (Houser et al., 2018). Offshore
bathymetric depth profiles were extracted every 1 m from a National Geophysical Data Center (NGDC) coastal relief model (CRM; Fig. 1).

Information about the subsurface framework geology of the coast was derived from a ~100 km alongshore EMI survey (Weymer et al., 2016; Wernette et al., 2018). EMI works by inducing a primary electromagnetic field in the subsurface half-
space and measuring the deformation (i.e. response) of a secondary current. From the secondary field deformation, it is possible to compute the apparent conductivity of the half-space at a specific frequency. While the apparent conductivity is influenced by a multitude of factors (Huang and Won, 2000; Huang, 2005), recent fieldwork suggests that hydrology has a minimal influence on the subsurface conductivity at PAIS at broad geographic scales, relative to the influence of stratigraphic and lithologic variation. A series of piezometer shore-normal transects were collected in fall 2016, which indicated that sand was
dry within the first 2 meters of the surface along the back beach. Since the EMI surveys were collected along the back beach, the piezometer measurements support the use of EMI as a proxy for the subsurface framework geology. Previous research confirms the location of several paleo-channels based on EMI surveys (Wernette et al., 2018; Weymer, 2016), while the current paper aims to determine the alongshore influence (direction and scale) of the paleo-channels.

## 2.3    Statistical Modelling of Spatial-Series

Previous research demonstrates that island morphology and framework geology can be spatially variable at multiple scales alongshore (Weymer, 2012, 2016; Wernette et al., 2018; Weymer et al., 2015a; Lentz and Hapke, 2011; Schwab et al., 2013; Hapke et al., 2016); however, previous approaches utilized models unable to identify spatial lags that may occur given





alongshore sediment transport gradients. Since the goal of this paper is to evaluate short- and long-range dependencies (SRD and LRD, respectively) of island morphology and framework geology and to test whether there is directional dependence in island morphology, the current study requires a statistical model capable of accounting for SRD and LRD. While fractal Gaussian noise (fGn) and fractal Brownian motion (fBm) models can model the SRD, both are unable to model the LRD of a

series because both models are limited to two parameters (fGn: range and standard deviation; fBm: variance and scaling). Therefore, we used an autoregressive fractionally-integrated moving average (ARFIMA) model to capture the LRD of a data series.

ARFIMA models may be considered a special case of autoregressive moving average (ARMA) models that have been most

widely applied in predicting financial market behavior; however, it is possible to analyze spatial data series by substituting space for time. The most significant advantage of ARFIMA models over ARMA, fGn, and fBm models is its potential to account for autoregressive (AR) relationships, LRD, and moving average (MA) relationships simultaneously through fitting $p$, $d$, and $q$ parameters, respectively. Many ARFIMA models utilize all three parameters simultaneously to describe a data series, although it is possible to isolate the influence of AR, LRD, or MA within the data in order to better understand more

specifically how the data is structured (Fig. 2). By isolating one of the three parameters, it is possible to distinguish the degree to which LRD influences a data series, independent of any SRD influence. This ability to distinguish and isolate LRD from SRD is unique and represents the most significant reason that ARFIMA models were used to test for directional dependencies in coastal geomorphology.

The $p$ and $q$ parameters provide information about SRD structures within the data series, representing AR and MA, respectively. Data series modelled with high $p$ values are those where the data value at particular location is dependent on the trend in nearby values. For example, large jetties or groins can affect the overall alongshore sediment transport (Fig. 2a and 2b), trapping sediment on the updrift side of the structure and starving downdrift areas of beach sediment. Alongshore beach-dune metrics, such as beach volume, provide valuable information about the alongshore influence of the coastal engineering

structures. Using an ARMA model to characterize the data series, we would find that $p$ values are very high adjacent to the jetties and decrease moving away from the structure (Fig. 2a). This simple AR relationship between the structures and beach volume is effectively represented by the $p$ parameter because this relationship is relatively localized to either side of the structures and the data series does not extend for several kilometers alongshore. Moving beyond the accumulated sediment on the updrift side or shadow on the downdrift side of the jetties, $p$ parameter values decrease. It is important to note that the $p$

parameter is useful for modelling localized AR relationships; however, given a more complex and/or substantially larger data series, the $p$ parameter is less likely to capture directional trends simply due to the increased "noise" inherent with larger data series. In other words, the AR relationships may become obfuscated with increasingly large and/or complex data series.




Data series modelled with high $q$ values also exhibit strong local dependence, although the data value at a particular location is dependent not on localized directional trends, but, on the average of nearby values (*i.e.* moving average). For example, assuming a groin field is effectively able to trap sediment and build a stable beach, the influence of these structures on beach volume can be effectively captured by the $q$ parameter (Fig. 2b). The $q$ parameter values for beach volume are much higher

within the groin field than outside of the field because the beach volume is being influenced by sediment trapped updrift and downdrift of a specific point. Similar to the $p$ parameter, it is important to note that the effectiveness of using $q$ parameter values to identify MA relationships decreases with increasing data series complexity and/or length. MA relationships are less evident in larger or very complex series simply because of the "noise" within the larger data series.

Unlike ARMA models which only utilize the $p$ and $q$ values, ARFIMA models include an additional $d$ parameter that can vary fractionally and provides information about the degree to which values within the series are dependent on all other values in the series, not simply localized effects (*i.e.* moving average and autoregressive). This $d$ parameter makes ARFIMA particularly well suited for modelling series with broad-scale dependencies (Fig. 2c). In the case of coastal geomorphology, $d$ parameter values may be particularly useful for identifying the influence of very broad-scale influencing factors, such as framework

geology (Weymer, 2016; Weymer et al., 2018).

ARFIMA modelling in the geosciences remains relatively unexplored, despite its potential for better understanding spatial and temporal patterns of variability in complex datasets. While previous research demonstrated that ARFIMA modelling can provide insight into long-range dependence patterns in alongshore barrier island surface and subsurface morphology at discrete

scales (Weymer, 2016; Weymer et al., 2018), the current paper expands the ARFIMA approach to analyse alongshore morphometrics at all scales along the entire length of spatial data series. In other words, while previous research utilized arbitrary alongshore lengths and locations to characterize LRD along PAIS, the current paper assesses LRD at all alongshore length scales along the entire length of PAIS. In this sense, the current paper presents a new approach to assessing how LRD changes alongshore and interprets these changes with respect to coastal processes and barrier island evolution.

In this paper, the effects of LRD within each spatial data series was isolated using a 0, $d$, 0 ARFIMA model. Each ARFIMA model was fit using the fracdiff package Fraley et al., 2012 in R (R Core Team, 2016), where the $p$ and $q$ parameters were set equal to 0. Setting both $p$ and $q$ parameters to 0 eliminates the short-range autoregressive and moving average terms from the fitted models. Each surface, subsurface, and bathymetric spatial data series contains 96,991 measurements in total. Each spatial

series was divided into ~250 unique computational windows, corresponding to alongshore length scales, ranging from two observations (2 m alongshore length scale) to the entire 96,991 observations (96,991 m alongshore length scale). While the number of computational windows can be decreased, or increased, it is important to note that the ARFIMA modelling process is computationally intensive. Increasing the number of computational windows would provide more detailed information about the structure of the dataset but would significantly increase the computing power required to fit the models. Decreasing the



number of computational scales would decrease the computing power required and speed up the computations; however, it would become more difficult to resolve the scales at which the structure breaks down. The range of computational windows could also be adjusted to a specific range, depending on the objectives of the research. At each scale the computational window is moved along the dataset and the appropriate *d* parameter is computed. The fitted *d* parameter is then assigned to the center

of the window at the corresponding length scale. Repeating this process for each alongshore length scale yields a matrix of values, where the row corresponds to the alongshore length scale of the data subset used to compute the *d* parameter, and the column represents the alongshore location of the center of the computational window. This matrix can be plotted similar to a wavelet plot to examine spatial patterns of LRD throughout the entire dataset at all length scales.

**2.4    Interpreting LRD Plots**

Figure 3 represents a sample LRD plot using a 10-km alongshore portion of PAIS dune height, where the x-axis represents the alongshore position or space (in meters) and the y-axis represents the alongshore spatial scale (in meters). Plots are oriented by latitude on the x-axis, from south (left) to north (right). In this paper, all plots utilize a color ramp from blue to red, where blue hues represent smaller *d* parameter values and red hues represent larger *d* parameter values. Given this color scheme,

locations or segments of the data lacking LRD are likely to appear as 'flares' or flames. Each of the flares, such as the flare at location A, represent the scale and areas of the dataset where LRD begins to break down in favor of SRD. LRD dominates at a particular location at a broad spatial scale (indicated by red hues) and becomes less influential as the spatial scale becomes increasingly finer (indicated by the transition from red to yellow to blue hues). In the case of the flare at location A (Fig. 3) we can see that the dune height series exhibits strong LRD at scales broader than ~20 km alongshore. This suggests that dune

height at location A is related to adjacent values down to ~10 km on both sides of A. Morphology at scales finer than ~20 m is more locally dependent. In this respect, ARFIMA represents an approach to determine the limiting scale to self-similarity.

Depending on the structure of the morphology and/or geology, it is feasible that the LRD may not appear to be symmetrical. Long-range dependence is asymmetric at location B, where the LRD begins to break down more rapidly to the right side of

the plot than the left. While the physical interpretation of a LRD plot depends on the variable, asymmetric flares can be broadly interpreted as areas where the variable is more locally dependent on the surrounding values at the scales and in the direction that the flare is oriented. In the case of flare B, dune height is more dependent on adjacent values to the north up to ~39 km alongshore. Asymmetries in the LRD plots can provide valuable information about the underlying structure influencing the variable of interest.

**3    Results**

The shoreline change LRD plot exhibits the greatest LRD values along the length of PAIS (Fig. 4b). Most flares present in the shoreline change LRD are at relatively fine spatial scales, shorter than a few kilometers. Peaks in the shoreline change LRD




plot are very narrow, suggesting that the long-term shoreline change is dominantly dissipative with only minor undulations due to localized coastal processes, consistent with Lazarus et al. (2011) who demonstrated that broad-scale and long-term shoreline change is dissipative. Waves impacting the coast can erode sediment from one area and transport it to another area, resulting in undulations in the shoreline orientation. Since long-term shoreline change is the result of cumulative daily wave

processes eroding undulations in the shoreline shape and dissipating any short-term undulations, fine-scale variations in the nearshore bathymetry, such as nearshore bars and troughs, can affect patterns of erosion and deposition along the coast over longer periods of time (Hapke et al., 2016). Therefore, it follows that the long-term shoreline change LRD plot would exhibit a large amount of LRD.

Beach width LRD is more variable than shoreline change (Fig. 4c), with the least amount of variability concentrated in the southern third of the island. Flares in the southern third of PAIS are likely present because transverse ridges in the nearshore bathymetry affect localized wave refraction patterns, thereby influencing fine-scale patterns in beach morphology. Patterns in the beach morphology in southern PAIS are likely more localized because the incoming wave energy is refracted around the transverse ridges, which impacts sediment transport gradients along this part of the island. Any variations in beach morphology

are more locally influenced by relatively closely spaced transverse ridges (~0.8 km to 1.5 km alongshore spacing), resulting in broad-scale LRD along southern PAIS.

The central third of PAIS beach width is characterized by several significant flares in LRD, with many of the strongest flares adjacent to infilled paleo-channels previously identified by Fisk (1959) (Figs. 4c and 5a). The scale at which LRD transitions

to SRD is at the broadest alongshore length scales proximal to Baffin Bay and this threshold decreases in scale to the north (Figs. 4c and 6a). Given a dominant southerly alongshore current during island development in the Holocene (Sionneau et al., 2008; Anderson et al., 2016) and corresponding southerly sediment transport gradient, patterns in the beach morphology LRD plot suggests that the paleo-channels are asymmetrically influencing beach morphology. It is plausible that paleo-channels acted as sediment sinks during barrier island formation. Simms et al. (2010) presented seismic profiles extending from north

to south across the ancestral LOSP Creeks, which exhibit a series of onlapping reflectors on the northern edge of the seismic profiles. These onlapping reflectors are indicative of deposition on the northern edge of the paleo-channel, and support the hypothesis that alongshore spit development occurred within the LOSP Creeks paleo-channel. The beach north of the large paleo-channel identified by Fisk (1959) would have been nourished by sediment discharged from the ancestral LOSP Creeks, now forming Baffin Bay. Similarly, the beach north of the ancestral LOSP Creeks paleo-channel may have been nourished by

sediment from the ancestral Nueces River. In this way, beach morphology updrift of the large paleo-channels would impact beach morphology within and south of the large paleo-channels.

Alongshore LRD in the dune crest elevation and dune height varies similarly to beach width LRD along PAIS (Figs. 4e, 4f, 5b, 5c, 6b, and 6c). The southern third of PAIS is characterized by LRD-SRD transitioning at finer alongshore length scales



than the northern two-thirds of the island, as indicated by the flares in the dune height LRD plot (Figs. 4e and 4f). The most significant flares are proximal to the ancestral LOSP Creeks paleo-channels dissecting central PAIS and the ancestral Nueces River paleo-channel extending into Baffin Bay (Fig. 6). Given that the dominant alongshore sediment transport gradient is from north to south and that the beach morphology exhibits an asymmetric LRD to the north of the large paleo-channels, it

follows that LRD and SRD patterns in dune morphology would exhibit similar asymmetry to beach morphology.

The transition from dune height LRD to SRD occurs at the largest scale, i.e. approximately at 35 km alongshore length scales (Figs. 4f and 6c). This maximum occurs at the southern edge of the ancestral LOSP Creeks paleo-channel, adjacent to Baffin Bay (Fig. 6c). The alongshore length scale can be interpreted as the alongshore distance that the paleo-channel affected wave

refraction patterns and sediment distribution along the beach, ultimately affecting sediment supply to develop larger dunes. It follows that paleo-channel influence on dune crest elevation and dune height would be asymmetric, with greater LRD to the north of the paleo-channels, assuming paleo-channels inhibited southern alongshore sediment transport and starved the beach downdrift. The wide beach updrift of a paleo-channel represents a larger sediment supply and greater fetch for aeolian transport and dune growth and is consistent with peaks in dune height identified by Wernette et al. (2018).

Island width exhibits the greatest alongshore variability in LRD of all island and framework geology morphometrics (Fig. 4g). Areas of short dunes are likely to be overtopped during a storm, transporting sediment to the landward margin of the island. Waves and currents along the landward margin of the island erode the washover fans and redistribute sediment along the island. In this sense, the island width at one location is directly influenced by sedimentation patterns along the adjacent parts

of the island. Undulations in the Gulf of Mexico shoreline are smoothed out over the long-term, thereby reducing the likelihood that patterns in island width are solely caused by shoreline change patterns. This repeat washover, followed by sediment redistribution along the backbarrier shoreline, represents the mechanism that barrier islands can transgress landward and keep up with sea level rise. The island width LRD plot demonstrates that island width is dependent on broad- and fine-scale patterns of change.

Bathymetric depth profiles at 2-km and 4-km offshore exhibit substantial LRD at broad scales but breaks down at scales finer than ~15 km alongshore (Figs. 4h and 4i). Long-range dependence breaks down at larger alongshore length scales in the 2-km bathymetry, compared to the 4-km bathymetry. Since modern coastal processes continue to affect alongshore sediment transport, large undulations in the bathymetry are smoothed out over time by sediment redistributed along the coast. Finer

scale variations in the modern nearshore bathymetry occur at similar spatial scales as previously identified at PAIS (Wernette et al., 2018). The 2-km bathymetric profile LRD breaks down at broader spatial scales than the 4-km bathymetry (Figs. 4h and 4i). This suggests that localized variations in coastal processes manifest in the nearshore bathymetry closer to the shoreline. Wave shoaling and breaking will erode and deposit sediment along the coast, impacting bathymetric structure closer to the shoreline.





Subsurface apparent conductivity exhibits substantial LRD along the entire length of PAIS (Fig. 4a). The substantial LRD along much of the island supports previous work by Weymer (2016) and Weymer et al. (2018), which demonstrated that subsurface framework geology exhibits LRD at discrete locations and alongshore length scales. Patterns in the subsurface

5  framework geology LRD plot demonstrate that the framework geology is self-similar at broader scales, and that this structure varies very little alongshore and with scale. The large LRD values at broad spatial scales (Fig. 4a) demonstrate that the paleo-topographic structure is trending towards a homogenous surface over very broad spatial scales. Since the framework geology reflects the paleo-topography and the modern barrier island surface is dissipative at very broad scales, based on large LRD values at broad scales in the modern barrier island morphology, it follows that the framework geology is dissipative.

## 4    Discussion

Dune height is an important morphometric to examine the influence of framework geology on barrier island morphology, since initial patterns in dune height and dune crest elevation can persist through time (Houser, 2012; Weymer et al., 2015b; Lazarus, 2016) and determine the response of a barrier island to storms (Sallenger, 2000). Areas of tall dunes are more likely to limit

washover and inundation during a storm, and instead be partially eroded from the dune and deposited on the beach and nearshore (Sallenger, 2000; Houser, 2012). Following the storm, sediment deposited in the nearshore is available for beach recovery through nearshore bar migration and welding. Onshore winds can transport sediment inland (i.e. from the beach to dune) following a storm, promoting dune recovery and development. Conversely, areas with shorter or no dunes are more likely to be overwashed or completely inundated, resulting in the net landward transportation of sediment to the backbarrier.

Since dune sand is not deposited in the nearshore or along the beach during the storm, sediment is not available for nearshore, beach and, eventually, dune recovery. In this way, variations in dune height and dune crest elevation are likely to persist through time by directly affecting patterns of overwash and represent a control on patterns of coastal resiliency and shoreline change. Identifying processes that set up modern patterns in dune morphology provides valuable insight into how the barrier island formed and how it continues to be influenced by the framework geology. Since dune height and development is partially

a function of beach width, it follows that beach width is a valuable morphometric to evaluate for patterns of LRD and SRD.

As noted, flares in the LRD plots are interpreted as areas where the morphometrics are more locally dependent on the adjacent values. Since flares in the LRD plots of surface morphometrics are most pronounced adjacent to the infilled paleo-channels and decrease to the north (Figs. 4, 5, and 6), this spatial correlation supports the hypothesis that the modern barrier island

morphology was influenced by variations in the framework geology. Paleo-channels along PAIS range in scale, with the smallest channels only ~13 m below the modern surface and the deepest and widest channels ~50 to ~64 m deep. Regardless of the paleo-channel dimensions, patterns in the LRD plots demonstrate that paleo-channels affect the nearshore bathymetry and modern island morphometrics asymmetrically and decrease in minimum alongshore scale to the north. Beach and dune




morphology updrift of a paleo-channel directly affects sediment available for areas of the beach downdrift. Given that a paleo-channel would have acted as a sediment sink for excess sediment transported alongshore during sea level transgression, it follows that LRD values would remain high at fine spatial scales updrift of the paleo-channel locations (Figs. 5 and 6).

The current paper is in agreement with previous research that demonstrates barrier island morphology is dissipative at broad spatial scales (Wernette et al., 2018; Lazarus et al., 2011). Long-range dependence is significant at very broad spatial scales in all island morphometrics except for island width. Previous research also demonstrates that rhythmic undulations and isolated paleo-channels can influence short-term shoreline change patterns (McNinch, 2004; Schupp et al., 2006; Lazarus et al., 2011) and beach and dune morphology (Houser et al., 2008; Houser and Barrett, 2010). This paper presents new information

supporting the hypothesis that paleo-channels in the framework geology asymmetrically influence barrier island geomorphology and that the scale of influence is ultimately limited. This asymmetry is likely caused by paleo-channels acting as sediment sinks for sediment transported south by a prevailing southerly alongshore current during barrier island formation.

The alongshore distance that variations in the framework geology influence beach and dune morphology is dependent on paleo-

15 channel scale and orientation, relative to the average shoreline orientation. Long-range dependence plots of beach and dune morphometrics suggest that beach and dune morphology within the largest paleo-channel dissecting the island, the ancestral LOSP Creeks, was influenced by beach and dune morphology up to 25 km north of the channel edge (Figs. 4c, 4d, 4e, 4f, 5 and 6). The large paleo-channel identified by Fisk (1959) is slightly smaller in scale than the paleo-channel forming Baffin Bay; however, the large Fisk (1959) channel intersects the coast at an oblique angle. Since the channel dissects PAIS at an

20 oblique angle, the influence of this channel is more apparent on beach morphology than dune morphology. An oblique channel would have required more sediment to fill than a shore-normal channel. Subsequently, a wide beach and dunes would begin to form in the shore-normal paleo-channel before the oblique paleo-channel. For an oblique paleo-channel the volume of sediment entering the channel would likely have been insufficient to build a wide beach to supply sediment for significant dune growth.

Paleocurrents during the Holocene were predominantly from north to south (Sionneau et al., 2008), which would have set up a southerly alongshore sediment transport gradient. Sediment transported from north to south along the coast would have nourished beaches updrift (i.e. north) of the channel. Consequently, nourished beaches updrift of the paleo-channel had a greater sediment supply and increased fetch for aeolian transport inland to promote large dune development (Bauer et al., 2009;

Bauer and Davidson-Arnott, 2002). While beach nourishment and dune growth continued updrift of the channel, excess sediment entering the channel was deposited along the updrift edge of the channel. Deposition on the updrift edge was caused by the increased accommodation space within the channel. Increasing the area that the alongshore current flows through (i.e. transitioning from a confined alongshore current to an open channel), while maintaining the alongshore current discharge, resulted in a decreased flow along the northern edge. Reducing alongshore current velocity caused sands to be deposited along



the northern edge of the channel, while finer particles are transported farther into the channel and funneled offshore through the channel outlet. Given enough time, this preferential deposition would have built a spit into the channel. Sediment trapped in the paleo-channel would be unavailable to the beach downdrift. The closest modern analogy to this alongshore sedimentation process is the formation and evolution of an alongshore spit forming a baymouth bar, where river valleys can become cut off

by the elongating spit and build large dunes on the updrift side.

Directional dependencies in beach and dune morphology, initially set up by the interaction of framework geology with a dominant southerly alongshore current, persist through time due to preferential washover reinforcing pre-existing alongshore variation in dune height. Areas of the island with limited or no dune development are preferentially overtopped by elevated

water levels during a storm. Conversely, areas with taller dunes resist storm washover/inundation and recover more rapidly following a storm. Alongshore variations in the barrier island morphometrics, such as dune height, persist through time because these patterns are re-enforced by episodic washover of small dunes during storms.

The apparent disconnect between long-term shoreline change and framework geology is due to the cumulative influence of

waves continuously interacting with the coast. This disconnect is further highlighted by the lack of storms impacting PAIS. Long-term shoreline change rate is the cumulative result of waves moving sediment along the coast on a daily basis, while short-term variations in shoreline position caused by storms are feasible. It is unlikely that short-term variations in PAIS shoreline position are caused by storms because PAIS has not been significantly impacted by a storm since Hurricane Bret in 1999. Any short-term undulations in shoreline position are likely to disappear over longer-time scales, especially since no

storm has hit the island to cause significant localized shoreline erosion. Therefore, the long-term shoreline change rate LRD (Fig. 4b) is unlikely to exhibit substantial variation alongshore. Beach, dune, and island morphology do show significant variation in patterns of LRD along PAIS (Figs. 4c, 4d, 4e, 4f, 4g, 5, and 6) because the initial barrier island morphology was set up by the framework geology. Predicting future changes to barrier island geomorphology requires a comprehensive knowledge of how the framework geology affected initial variation in the beach and dunes.

Understanding how the framework geology influences barrier island geomorphology has important implications for understanding how barrier islands are likely to recover following a storm or series of storms. While many models of barrier island recovery focus on spatio-temporal models of change, Parmentier et al. (2017) demonstrated that spatial autocorrelation outperformed temporal autocorrelation (e.g. "space-beats-time", SBT) when predicting the recovery of vegetation following

Hurricane Dean. Since vegetation recovery and dune geomorphic recovery are related (Houser et al., 2015), it follows that spatial autocorrelation in beach and dune features is essential to predicting future changes to barrier island geomorphology. The current paper supports the conclusions of Parmentier et al. (2017) by demonstrating that spatial variations in the framework geology directly relate to alongshore variations in beach and dune morphology (Figs. 5 and 6). In context of SBT theory, results of the current paper support the hypothesis that spatial variations in the framework geology (i.e. 'space') control barrier



island evolution (i.e. 'time'). Accurately predicting future barrier island change is predicated on comprehensively understanding what processes influenced its initial formation and what processes continue to influence island morphology.

Given that framework geology influences beach and dune morphology along the coast, the methods and results of this paper represent an opportunity for managers to improve coastal nourishment projects. Sediment budget imbalances set up by the framework geology dictate long-term barrier island trajectory. Utilizing ARFIMA models to evaluate the alongshore beach and dune morphology can provide valuable insight into the coast is likely to change naturally in the future. To reduce waste by coastal nourishment, future projects should seek to first comprehensively understand how the paleo-topography of an area continues to affect coastal processes and morphology. By understanding the long-term influence of framework geology, coastal nourishment projects can more effectively balance how a project focused on the near-future coastal morphology with long-term natural changes. Although there is no single solution to managing coastal resources, effective long- and short-term management of coastal resource should seek to balance societal pressure with natural long-term behavior to minimize economic and environmental loss.

## 5 Conclusion

This paper quantitatively demonstrates that variation in the framework geology influences patterns of beach and dune morphology along a barrier island. Understanding what controls beach and dune morphology and barrier island development is integral to predicting future changes to barrier island geomorphology and island transgression caused by storms and sea level rise. Storm impact and barrier island transgression patterns are controlled by beach slope, dune height, and wave run-up. Given a persistent alongshore sediment gradient during the Holocene, paleo-channels in the framework geology at PAIS likely acted as sediment sinks during island development. While wide beaches and, subsequently, large dunes are nourished with sediment updrift of the channel, excess sediment can become trapped in the channel. These channels trap sediment, starving sediment from downdrift portions of the coast. The result of this asymmetry in sediment supply is that large dunes occur updrift of the paleo-channel and small dunes occur downdrift of the paleo-channel. Effectively managing a barrier island underlain by a variable framework geology should seek to balance short-term societal pressures in context of long-term natural change (*i.e.* framework geology).

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

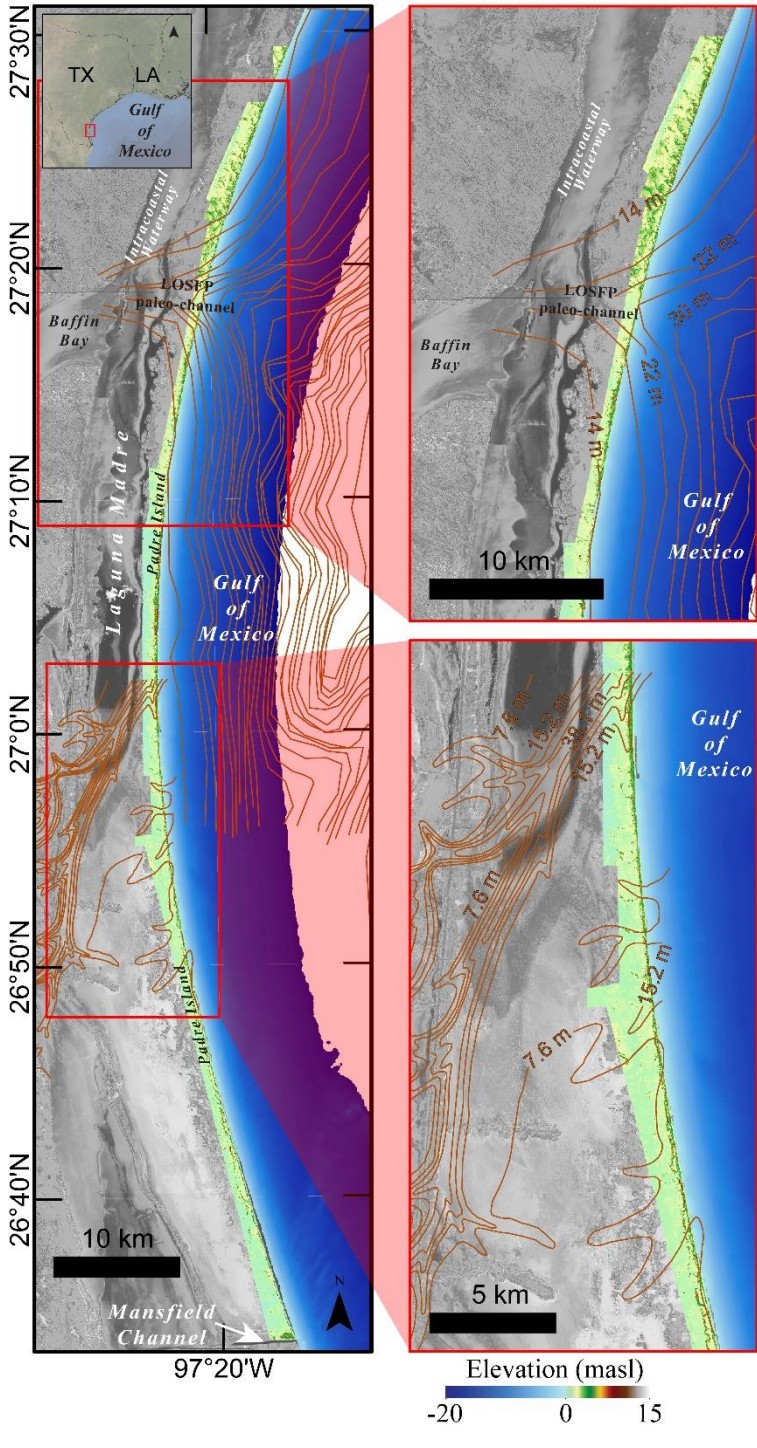

**Figure 1:** Padre Island National Seashore represents an ideal location to test topobathy DEM with Pleistocene paleo-surface contour lines from Fisk (1959) and MIS II paleo-surface contour lines from Anderson et al. (2016).





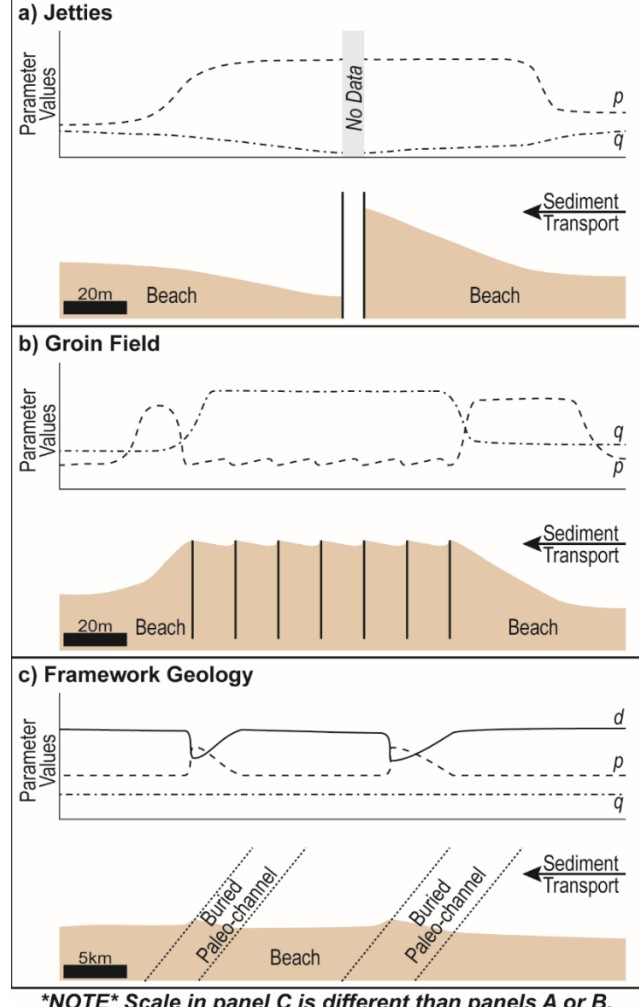

**Figure 2:** Sample beach-dune alongshore data series and ARFIMA model parameters ($p$ = AR; $d$ = LRD; and $q$ = MA) for three coastal geomorphology scenarios. (a) Jetties trap sediment on the beach updrift side and starve the downdrift beach of sediment (see Ocean City, Maryland, USA), resulting in increased AR values on either side of the jetty. (b) Groin fields can

5 trap sediment between the groins within the field, while starving the downdrift beach of sediment. In this case, beach volume at a particular location within the field can be modelled as the MA of adjacent beach volume measurements. Outside of the groin field, beach volume may increase/decrease, resulting in increased AR values and decreased MA values. (c) Framework geology, such as infilled paleo-channels, influences coastal geomorphology on broader spatial scales (see oblique gravel ridges in the Outer Banks, North Carolina, USA) is much more likely to appear in the LRD values. While coastal morphology at

10 broad-scales is influenced by the entire data series, sediment transport gradients can be influenced by more localized processes, resulting in an inverted trend with the AR component. The degree to which a particular point is influenced by the entire data series at a particular scale can be modelled and plotted using the LRD parameter.



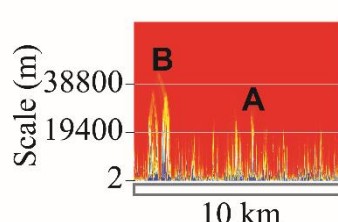

**Figure 3:** Example LRD plot using alongshore dune height at PAIS. The y-axis represents the alongshore length scale (in meters), and the x-axis represents the alongshore location. LRD is persistent at greater alongshore length scales at location B than location A. Additionally, location B is asymmetric, which may suggest a directional dependence in the data series.







**Figure 4:** Long-range dependence plots of alongshore morphometrics: (a) 3 kHz apparent conductivity, (b) shoreline change rate (end-point rate), (c) beach width, (d) dune toe elevation, (e) dune crest elevation, (f) dune height, (g) island width, (h) bathymetric depth profile at 2-km offhsore, and (i) bathymetric depth profile at 4-km offshore. All LRD plots are aligned with the map below, based on latitude. Previously documented variability in the framework geology is indicated by the contour lines representing the Pleistocene (*i.e.* MIS II) paleo-surface (Anderson et al., 2016;Fisk, 1959).



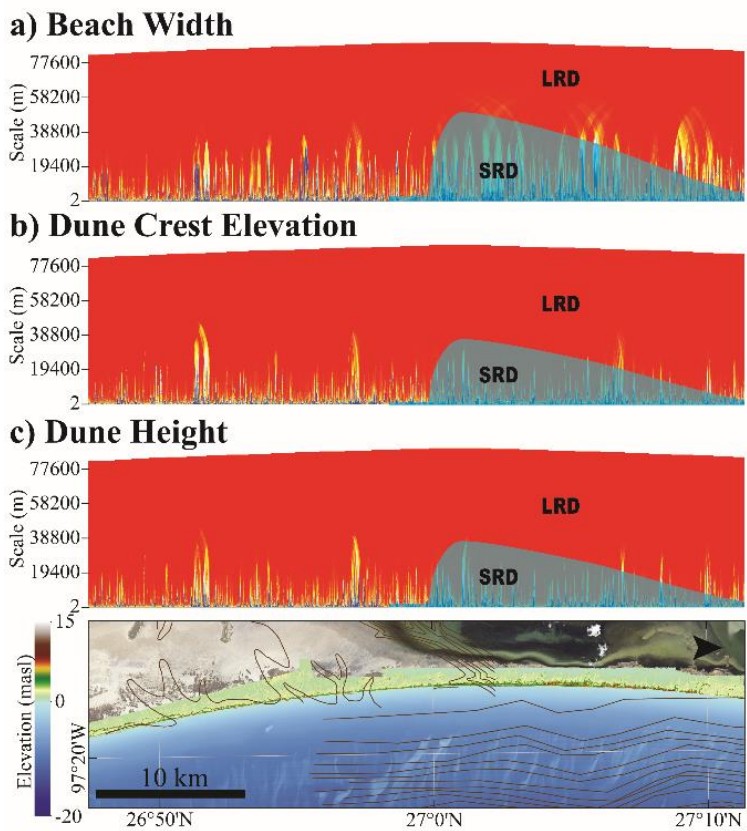

**Figure 5:** LRD plots of (a) beach width, (b) dune crest elevation, and (c) dune height for central PAIS, where Fisk (1959) identified a series of relict infilled paleo-channels dissectting the island. The scale at which LRD breaks down in favor of SRD is greatest at the southern edge of large paleo-channels, and this scale gradually decreases to the north. Smaller paleo-channels do not appear to be as influential to the modern beach and dune morphology, suggesting that small channels may not have as significant an influence as larger channels.





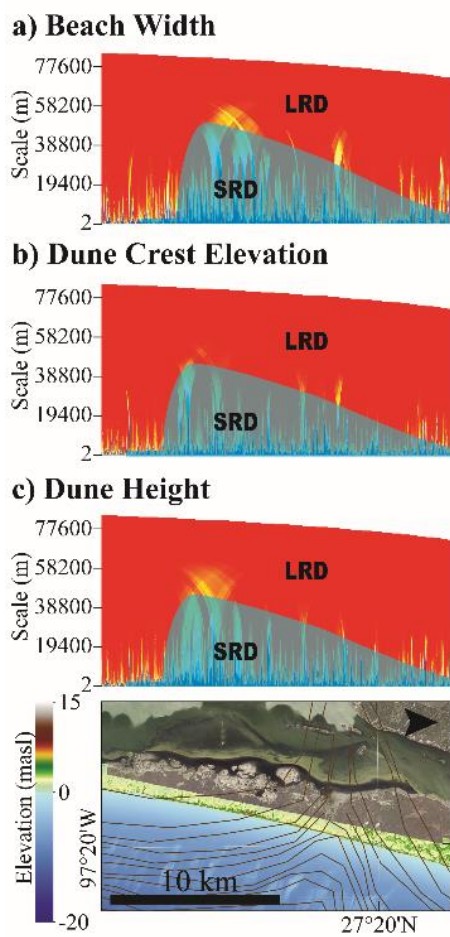

**Figure 6:** LRD plots of (a) beach width, (b) dune crest elevation, and (c) dune height for PAIS adjacent to the ancestral LOSP Creeks, forming the modern Baffin Bay. LRD breaks down in favor of SRD at the largest scales at the southern edge of the previously identified paleo-channel. The scale at which LRD breaks down to SRD decreases gradually to the north of the channel, suggesting that the paleo-channel asymmetrically influenced beach and dune morphology.