# Peer review of "Directional dependency and coastal framework geology: Implications for barrier island resilience"

_Earth Surface Dynamics, 2018_

## Referee Comment (RC1) · A. Cooper (Referee) · 7 Jul 2018

Using novel approaches appropriate to the task, and based on an impressive dataset, This paper presents important new insights into the link between geological framework and barrier island morphology and behaviour. Central to the paper is that Padre Island is one of the few places worldwide that adequate data exist to draw the conclusions presented here. In most parts of the world therefore, there is thus an important preliminary step of mapping the subsurface geologic framework before any such dependencies can be identified, let alone incorporated into any practical management scheme. One of the key outcomes of this work should be to highlight the need for such data in order to improve coastal management decisions.

At first I was concerned that this was the same paper I had reviewed for another journal. However, while it uses the same dataset and similar approaches, it does differ in its investigation of LRD AND SRD and has important new findings in this regard. However, the authors need to be explicit early in the paper about how this work departs from or adds to their previously published findings. In the current ms this only happens at line 255. This may also require appropriate changes to the title. Several messages are contained in the paper, some of which overlap to a greater or lesser extent with the authors' previously published work. Consequently, a deliberate focus on the key departure of this paper is essential.

There are several points that require further explanation and/or clarification. For example: what is implied by "asymmetry" as opposed to irregularity is not immediately clear; there is an apparent contradiction between line 170 and 310 regarding shoreline change data; the text refers to LRD and SRD being aims of the paper and yet up to that point LRD was the overwhelming focus (as implied in the title).

In several places improvements in syntax are needed. These are indicated in the attached pdf file with notes attached. Also, a review paper was published in Global and Planetary Change, probably after submission of this ms, that considers the role of geological factors on mesoscale barrier behaviour. It should be cited.

Please also note the supplement to this comment:
https://www.earth-surf-dynam-discuss.net/esurf-2018-41/esurf-2018-41-RC1-supplement.pdf
* * *
[Figure]

**Supplement:**

**Long-range dependence in coastal framework geology: asymmetries and implications for barrier island resiliency**

Phillipe A. Wernette*[1,2], Chris Houser[1], Bradley A. Weymer[3], Mark E. Everett[4], Michael P. Bishop[2], and Bobby Reece[4]

[1]University of Windsor, Department of Earth and Environmental Sciences, Windsor, Ontario, Canada N9B 3P4
[2]Texas A&M University, Department of Geography, College Station, Texas, USA 77843
[3]GEOMAR Helmholtz Centre for Ocean Research Kiel, 24148 Kiel, Germany
[4]Texas A&M University, Department of Geology and Geophysics, College Station, Texas, USA 77843

*Corresponding author email: wernette@uwindsor.ca
*Corresponding author present address: University of Windsor, Department of Earth and Environmental Sciences, 401 Sunset Ave., Windsor, Ontario, Canada N9B 3P4

**Abstract**

Barrier island transgression is influenced by the alongshore variation in beach and dune morphology, which determines the amount of sediment moved landward through washover. While several studies have demonstrated how variations in dune morphology affect island response to storms, the reasons for that variation and the implications for island management remain unclear. This paper builds on previous research by demonstrating that the framework geology can influence beach and dune morphology asymmetrically alongshore. The influence of relict paleo-channels on beach and dune morphology on Padre Island National Seashore, Texas was quantified by isolating the long-range dependence (LRD) parameter in autoregressive fractionally-integrated moving average (ARFIMA) models. ARFIMA models were fit across all scales and a moving window approach was used to examine how LRD varied with computational scale and location along the island. The resulting LRD matrices were plotted by latitude to place the results in context of previously identified variations in the framework geology. Results indicate that the LRD is not constant alongshore for all surface morphometrics. Many flares in the LRD plots correlate to relict infilled paleo-channels in the framework geology, indicating that the framework geology has a significant influence on the morphology of PAIS. Barrier island surface morphology LRD is strongest at large paleo-channels and decreases to the north. The spatial patterns in LRD surface morphometrics and framework geology variations demonstrate that the influence of paleo-channels in the framework geology can be asymmetric where the alongshore sediment transport gradient was unidirectional during island development. The asymmetric influence of framework geology on coastal morphology has long-term implications for coastal management activities because it dictates the long-term behavior of a barrier island. Coastal management projects should first seek to understand how the framework geology influences coastal processes in order to more effectively balance long-term natural variability with short-term societal pressure.

Keywords: coastal geomorphology; beach-dune morphology; barrier island development; long-range dependence; spatial data analysis; framework geology

[revised manuscript text omitted]

---

## Referee Comment (RC2) · J. Miselis (Referee) · 8 Aug 2018

In this study, Wernette et al. examine spatial relationships between barrier island geomorphology and framework geology, specifically the presence or absence of subsurface paleo-channels. To make this comparison, they have: 1) extracted geomorphological metrics from topographic data along the extent of Padre Island National Seashore (PAIS), such as beach width, dune height, etc.; 2) used variations in apparent conductivity measured from an electromagnetic induction survey to identify the location of paleo-channels; 3) adapted an autoregressive fractionally-integrated moving average (ARFIMA) model to identify long-range spatial dependencies in each data series; and 4) identified similarities between barrier island and geology spatial data series as a proxy for a long-term process-based relationship. Using quantitative methods to identify spatial relationships between coastal morphology and framework geology is a necessary pursuit and it's one that these authors are well-equipped to take on.

In this paper, it's clear that the authors are very comfortable with statistics-based spatial modelling; Section 2.3, in which the statistical methods are introduced, is well-written, clear (even to someone unfamiliar with the specific technique), and one of the best parts of the paper. Unfortunately, the rest of the paper is not as successful as it could be. First, explicit distinctions between this work and Wernette et al., 2018 (Influence of a spatially complex framework geology on barrier island geomorphology, Marine Geology, 398, 151-162) should be made, particularly because both papers rely on the same dataset but use different spatial analysis techniques. Their stated purposes are very similar (compare P3, L11-13 of this paper to "The purpose of this paper is to examine the influences of framework geology on beach and dune geomorphology...where the framework geology is variable alongshore." from Wernette et al., 2018). There are differences, but those differences need to be better highlighted in this paper. The introduction, in particular, misses an opportunity to put this work into its own context. Potential hypotheses are stated in the introduction (P3, L11-13 and P3, L22-26), but the supporting text is too general. The authors are testing a specific feature in the geologic framework (e.g., paleo-channels) and I think a paper that focuses on hypotheses that explore the role these features play in coastal evolution may distinguish it from earlier work and ultimately, might be more successful.

Second, the interpreted physical meaning of the results of the statistical modeling could be better organized and supported. Section 3 could benefit from some reorganization that first states the trends observed in the ARFIMA modeling results. This could be followed by the authors' physical interpretation of those results and then a brief description of their reasoning for that interpretation. Having the results better organized might make the conceptual model for the interactions between barrier island formation and paleo-channel characteristics presented in the discussion (∼P12, L14-P13, L13) seem like less of a leap. However, even with a clearer results section, their conceptual model of channels as sinks of sediment during barrier island formation (which then results in smaller dunes and beaches, initially and now) needs more development. A schematic illustration of the process the authors describe in the context of the geologic evolution of the system (e.g., Weise and White, 1980), particularly sea level elevation, would be a great start. Finally, it's worth noting that their conceptual model relies heavily on the assumption that "initial patterns in dune height and dune crest elevation can persist through time." Later in the discussion, the authors go on to state that their study area hasn't been significantly impacted by a storm since 1999. Some discussion of the extent to which this assumption and ultimately their results would hold if a storm did impact the island and how their conceptual model might apply outside of the Gulf of Mexico would help to frame the scope of the audience for this contribution.

As it stands now, I cannot recommend this manuscript for publication. However, with some thoughtful revision, this paper would be an interesting contribution to a growing body of work by these authors and would be of interest to coastal geologists and geomorphologists and other scientists concerned with the long-term evolution of coastal systems and its impact on modern coastal processes.

Technical Comments: Manuscript

I've included line-by-line comments below to accompany the highlighted copy of the manuscript uploaded as a supplement.

P1, L14: What does this mean? Is there another way to describe this relationship? Since this is the primary distinction from previous work and it is used several times in the abstract, the meaning should be immediately clear and/or defined at first use.

P1, L19: Without reading the paper, the reader may not have the context for what this means. Use another word to describe what you mean…"peaks?"

P1, L20 and L23: Why not just "paleo-channels" here? The current phrasing is redundant.

P2, L3 and L5: Rewrite these two sentences to eliminate redundancy.

P2, L7: What is meant by "patterns of vulnerability?"

P2, L10-11: I think you mean "where VARIABILITY IN geologic structure can RESULT FROM variations in..., yes?

P2, L19-20: There are many other models that assume alongshore uniformity beyond the probabilistic ones listed here. Consider citing a few of those as well. Also, Long et al., 2014 is not a modeling paper...did you mean to cite something else?

P2, L28-30: Citation?

P2, L30-33: Try to summarize the contributions from Houser, 2012 more succinctly (e.g., in one sentence) so as not to lose the focus of the paragraph.

P3, L3-5: I do not understand this sentence. Reword.

P3, L6-9: Why is vegetation being brought up here? How does this sentence help you set up the scientific problem you're testing?

P3, L11-13: Regarding the stated purpose of this paper, how is this different than the purpose of your 2018 paper in Marine Geology, which states: "The purpose of this paper is to examine the influence of framework geology on beach and dune geomorphology at Padre Island National Seashore (PAIS), Texas, USA, where the framework geology is variable alongshore."? Use the introduction of this paper to make those distinctions explicit.

P3, L22-26: Here is the hypothesis! Right? Rewrite the introduction around this hypothesis?

P5, L3-5: First mention of economic forecast models! Ok, so what is the purpose of this paper? It sounds like it is to determine how paleo-channels have influenced beach and dune evolution. But this isn't a process-based study, so how are you going to determine "the how?" It seems more likely that what you are trying to do is to adapt an economic forecast model to explore the spatial relationships (Are they lagged spatially? Is there a scale-dependence?) between beach and dune morphology and the presence/absence of subsurface paleochannels.

P5, L27-28: Here the distinction between the 2018 paper and this paper is clearly stated. But I would argue that Wernette et al., 2018 did more than "confirm the location of several paleo-channels." (That's actually what some of your co-authors papers, Fisk, and Anderson et al. did, no?) It also established a spatial connection between the presence of those paleochannels and beach and dune morphology by applying several signal analysis techniques in space. Be careful to distinguish this work from previous work, particularly Wernette et al., 2018.

P5, L34-P6, L1: The relationship between spatial lags and alongshore sediment transport gradients in dictating spatial relationships between geologic features and coastal response is a really important point. It would be good to spend some time discussing this in the introduction. The idea that the effect of longer-term influences on cross-shore processes are going to be diffused by alongshore transport is definitely worthy of some hypothesis testing!

P6, L9-18: Great description of the reasoning behind applying this model.

P7, L13-15: "Identifying the influence of very broad-scale influencing factors"? Rewrite this sentence to clarify what you mean. It's probably a good idea to change "framework geology" to "subsurface paleo-channels" or "paleo-channels" to help maintain the focus of THIS paper.

P7, L18-24: The distinction between this paper and previous work is clear in this section. Nicely stated. Should the focus of the paper be on the application of this technique to coastal geomorphology, specifically the relationship between paleo-channels and beach/dune morphology? The relationships identified in previous work could be used as justification for this "experiment." In fact, some evaluation of this technique over (or in place) of others previously used would be appropriate somewhere in the paper. In other words, could you just start with this analysis rather than those done in Wernette et al., 2018 or are those a prerequisite for ARFIMA modeling?

P7, L29-31: Complete this discussion with an explanation of how you arrived at the decision to use 250 unique computational windows; why not 200 or 300? How much do you have to increase or decrease the number of windows before you experience large increases in computing power or lose the ability to resolve the breakdown in spatial structure, respectively?

P8, L33: Shouldn't the results section start with a description of plot 4a? Also, what is meant by "greatest LRD values?" Consider adding "as indicated by the dominance of red on the plot" or similar to keep the reader with you and reinforce the information from the previous section. Why doesn't apparent conductivity (Fig. 4a) show "the greatest LRD values?"

P8, L34-P9, L3: Here and throughout the results section, it might be better to state your observations from the plots followed by your "interpretation" of the physical meaning. Here you might say, "Peaks in the shoreline change LRD plot are very narrow, which we interpret to mean that long-term shoreline change is dominantly dissipative...". Use the discussion to reference papers that support your interpretations or point out where they are not consistent with existing literature.

P9, L11: Indicate on Fig. 4 where the island is divided into thirds (southern, central, and northern) so the reader can easily follow the text and identify the features to which you refer.

P9, L23-24: This is speculation. Move this to discussion where you can discuss your reasoning for this and support it with evidence from the literature.

[Figure]

P9, L33: How are dune crest elevation and dune height distinct from one another? Is there an advantage to using both rather than just one of these metrics?

P10, L16 & P10, L23-24: Follow the first sentence of this paragraph with your interpretation of what it means (the last sentence of the paragraph) and then go in to the details of the broad- and fine-scale patterns that drives your interpretation. This structure could be applied to the entire results section, particularly the next paragraph (P10, L26-34).

P11, L1-9: Either move this paragraph to the beginning of the results section to be consistent with the figure or change the figure to be consistent with the text. Also, the logic in this paragraph is a bit jumbled. Rewrite to follow the structure of observation→interpretation→reasoning.

P11, L4-7 (highlighted in green): Are there patterns in Fig. 4a? If so, they are not apparent in the figure. Or is it the lack of any patterns (e.g., red everywhere) that you're referring to? Also, the part of the sentence that states "this structure varies very little alongshore and with scale" seems to contradict your statements elsewhere in this paper and in Wernette et al., 2018 that framework geology varies alongshore. Clarify. Finally, what do you mean by "the paleo-topographic structure trends toward a homogenous surface at broad scales?" So, if you shrunk the alongshore length scale the trend would be different?

P11, L12-25: This paragraph seems an odd fit here. It seems as if it's defending the choice of barrier island metrics used in the paper, which, if necessary at all, might be better suited for section 2.2. If that's not the point of this paragraph, it should be revised for clarity.

P11, L30-31: The point about the depth of the paleo-channels is worth making earlier in the paper. If I understand what you're proposing correctly, the relationship that you're trying to tease out is the relationship between paleo-channels and barrier island morphology as the paleo-channels were infiling and (perhaps?) as the barrier island was forming. And that you are using the assumption that those dune heights/elevations initially established are what's driving the modern morphology now (see P11, L6-25), based on Houser, 2012; Weymer et al., 2015b; and Lazarus, 2016. This is quite the hypothesis, particularly since it would also assume that the barrier island had the same configuration as it does today (which is likely not the case). How can you better convince the reader that your hypothesis is supported by your analysis?

P12, L10-11: The use of "asymmetrically" here is confusing. I think what you mean is that paleo-channels and alongshore currents interact to drive asymmetries in their spatial relationship(s) to barrier island morphology, yes?

P12, L18-24: The end of this paragraph is jumbled. Rewrite for clarity. It seems that ultimately this is a discussion of sediment supply as the channel was infilling and the island/beach/dunes were forming. I think what you're trying to get at is that oblique channels would have taken more sediment to fill relative to shore-normal paleo-channels, thereby leaving less sediment available for building beaches and dunes?

P12, L26- P13, L12: This section would benefit from a schematic of the processes you describe. Where is sea level when these processes are ongoing? Is this consistent with published theories of coastal evolution for the area (a la Weise and White, 1980)?

P13, L3-5: Citation for the baymouth-bar analogy?

Technical Comments: Figures

Figure 2. Reference to "oblique gravel ridges" in caption of panel c is incorrect. Gravel was present in the troughs between sandy shore-oblique bars. Suggest a citation for Browder and McNinch, 2006 here instead.

Figure 3. This figure appeared to be ∼1.5 inches high by 2 inches wide in the submitted manuscript. Consider making this figure much larger so the reader can follow your interpretation of the results, particularly with regard to asymmetry. For example, is your interpretation of asymmetry due to what appears to be a "tail" that trails off to the left at the top of the peak at "B" or due to the concentration of darker blue toward the left of the bottom of the right peak under "B?"

Figure 4. It's very difficult to see the labels on the x and y axes in all of these plots. It could be separated into two figures, one with AC, Shoreline Change, 2km Bathy, and 4km Bathy and a second with all of the island metrics (BW, DTE, DCE, DH, and IW).

Figure 5. I don't understand the justification for the coloration of the northern half of these panels with the SRD polygon. What are you looking at to justify the northward scale decrease? In panel B, there's a peak about an inch in from the left that's taller than the peaks south of it which would seem to contradict the interpretation as I understand it.

Please also note the supplement to this comment:
https://www.earth-surf-dynam-discuss.net/esurf-2018-41/esurf-2018-41-RC2-supplement.pdf

**Supplement:**

[revised manuscript text omitted]

---

## Author Comment (AC1) · 12 Sep 2018

The authors would like to thank reviewers for comments which helped improve the manuscript significantly. Each reviewer comment is addressed in detail below, and the revised manuscript with track changes has been attached to the end of this document.

**Response to RC1**

One of the key outcomes of this work should be to highlight the need for such data in order to improve coastal management decisions.

> The authors believe that this is better articulated in the revised manuscript after addressing all of the following reviewer comments.

The authors need to be explicit early in the paper about how this work departs from or adds to their previously published findings. In the current manuscript this only happens at line 255. This may also require appropriate changes to the title. Several messages are contained in the paper, some of which overlap to a greater or lesser extent with the authors' previously published work. Consequently, a deliberate focus on the key departure of this paper is essential.

> The title, abstract, introduction, and discussion have been updated to distinguish this paper from previous papers.

L6 "asymmetrically": Is this really what you mean? Isn't simply "irregular" a better term?

> The text has been updated to clarify that the authors are referencing an 'irregular' framework geology with paleo-channels:
>
> *This paper builds on previous research by demonstrating that paleo-channels in the irregular framework geology can have a directional influence on alongshore beach and dune morphology.*

L9 "long-range dependence (LRD) parameter": Dependence of what on what?

> A more complete description of the meaning of LRD is not feasible in the abstract, given length requirements. However, the Methods section does contain this information.

L10 "(ARFIMA) models": What do they model?

> The text has been updated to clarify that ARFIMA models were originally developed for stock market economic forecasting:
>
> *The influence of relict paleo-channels on beach and dune morphology on Padre Island National Seashore, Texas was quantified by isolating the long-range dependence (LRD) parameter in autoregressive fractionally-integrated moving*

*average (ARFIMA) models, originally developed for stock market economic forecasting.*

L10 "all scales": Temporal and spatial?

The text has been updated to clarify that "~250 unique spatial scales" were used in the analysis, in combination with the moving window approach.

L16 "PAIS": If PAIS means "Padre Island National Seashore", it is a strange acronym.

PAIS is the official acronym for Padre Island National Seashore used by the U.S. National Park Service.

L19 "asymmetric": OK, so asymmetric means having a longshore gradient away from point sources. Is this correct?

This is correct. Text was updated to further clarify this potential point of confusion:

*The spatial patterns in LRD surface morphometrics and framework geology variations demonstrate that the influence of paleo-channels can be asymmetric (i.e. affecting beach-dune morphology preferentially in one direction alongshore) where the alongshore sediment transport gradient was unidirectional during island development.*

L23 "how the framework geology": In most cases it is first necessary to assess the framework geology.

Text was revised to include assessing the framework geology as a first essential step:

*Coastal management projects should first seek to assess the framework geology and understand how influences coastal processes in order to more effectively balance long-term natural variability with short-term societal pressure.*

L54: See also the review by Cooper et al. 2018. Global and Planetary Change.

The reference was added to support the importance of understanding how framework geology affects coastal geomorphology.

L65 "headlands along Fire Island, NY reflected in": (missing) "are"

The text has been updated to reflect this suggestion.

L75 "Lazarus et al. (2011) argued that": Give page number.

The page number was added to the in-text citation.

L80 "beach and dune morphology and the dissipative": (missing) "therefore"

This sentence has been split into two based on this and RC2's suggestions:

*This implies that variations in the framework geology, such as paleo-channels, do not influence long-term shoreline change, but, as noted, shoreline change is influenced by the alongshore variation in beach and dune morphology. The dissipative behavior of shoreline change does not negate the importance of framework geology.*

L100 "gradient during the time": (word choice) "that"

The text has been updated to reflect this suggestion.

L105-107: This sentence seems out of place and could be deleted. The several intermediate steps between identifying framework controls and coastal management would need to be described to render it a useful statement.

This sentence has been removed.

L126 "drown": (word choice) "drowned"

The text has been updated to reflect this suggestion.

L126 "relict dunes": Aeolian?

Presumably these were aeolian, but I am unable to locate any literature that definitively confirms that they were originally aeolian landforms.

L126 "approximately 80 km inland": Of the LGM shoreline?

Approximately 80 km inland of the LGM shoreline is correct. The text has been updated to reflect this clarification.

L134-135: The paleo landsurface and ravinement surface need not coincide. They are not synonymous.

We acknowledge that this is not always the case; however, in the case of PAIS, the ravinement surface is the best approximation we have of how the paleo-landscape may have looked. Depth to the MIS II surface (as mapped by Anderson et al., 2016) and depth to the 'Pleistocene ravinement surface' (from Fisk, 1959) roughly correspond along the island, further supporting our interpretation of the ravinement surface as representing the paleo-surface during MIS II.

L144 "complexities in the framework geology": More than that, the very fact that the framework geology has been measured, makes it an ideal location.

> The text has been updated to add that simply having an existing map of the framework geology is a substantial benefit to using PAIS for this study:
>
> *Complexities in the framework geology and modern island geomorphology, coupled with the fact that PAIS framework geology has already been mapped, make PAIS an ideal location to examine how framework geology influences barrier island geomorphology.*

L157-160: Just how much does it expand on these authors' previous work?

> The text has been updated to clearly describe how the current work expands on previous work by examining spatial lag distances in how paleo-channels affect beach-dune morphology:
>
> *The current paper expands on previous research by adapting economic forecast models to determine how paleo-channels in the framework geology have influenced beach and dune evolution and whether this influence is directional and scale-dependent. Identifying these spatial lags, their spatial scale(s), and their lag direction(s) is the first step toward integrating this information into morphodynamic prediction models.*

L169-171: How does this square with line 310 et seq.?

> The text has been updated to include shoreline change as a variable used in this paper:
>
> *Long-term shoreline change (1950-2007) was used in this analysis because fine-scale and shorter-term changes are unlikely to persist given that there has not been ample storm activity to continually force shorter-term variations in shoreline change at PAIS (Houser et al., 2018).*

L195 "SRD and LRD": The title only refers to LRD. Also, please define short vs long.

> A sentence has been added in section 2.3 to define SRD and LRD and the difference between them:
>
> *Short-range dependence includes localized relationships in the data series, such as autoregressive or moving average, there LRD is the dependence of values on all other data values within the data series, irrespective of trend or window size.*

L230 "may become obfuscated with increasingly large and/or complex data series": How does one know whether this has happened? Is there an optimum data density?

While it would be possible to isolate the autoregressive parameter, $p$, in a series of ARFIMA models where the data series length is gradually increasing, this analysis is beyond the scope of this paper. The current paper is focused on understanding broad-scale patterns in geomorphology through variations in the LRD (*i.e.* 'self-similarity') parameter, *d*.

L256-260: OK, this statement eases my concern about the overlap with the authors' previous work. To that extent, this point needs to be made much earlier and much more clearly in the paper.

The text has been updated to clarify the difference between this paper and previous publications:

*While wavelet decomposition can provide insight into relationships between two variables in the same location (Wernette et al., 2018), utilizing ARFIMA as a sliding window across multiple spatial scales can shed light on relationships that exhibit a lag in one or both directions. The advantage of this new approach is its application to examine alongshore influences of various natural and anthropogenic features (e.g. jetties, seawalls, groin fields, paleo-channels, and/or headlands) and identify their effective zone(s) of influence on coastal processes and geomorphology.*

L314 "consistent with Lazarus et al. (2011)": (missing) "the findings of"

The text has been updated to reflect this suggestion.

L371 "short dunes": What is a short dune? = low dune?

Variations on 'tall' and 'short' have been replaced by 'high' and 'low,' respectively. This change is throughout the manuscript.

L378 "mechanism that barrier islands": (word choice) "by which"

The text has been updated to reflect this suggestion.

L382 "broad scales but breaks down": (missing) "this"

The text has been updated to reflect this suggestion.

L408 "tall dunes": High and low are preferable in this context to short and tall.

Variations on 'tall' and 'short' have been replaced by 'high' and 'low,' respectively. This change is throughout the manuscript.

L409 "instead be partially": (word choice) "sediment is likely to be"

The text has been updated to reflect this suggestion.

L413 "areas with shorter": (word choice) "low"

Variations on 'tall' and 'short' have been replaced by 'high' and 'low,' respectively. This change is throughout the manuscript.

L480 "build large dunes on the updrift side": River valleys don't build dunes. What processes are actually happening?

The text has been updated to improve clarity:

*The closest modern analogy to this alongshore sedimentation process is the formation and evolution of an alongshore spit forming a baymouth bar, where river valleys can become cut off by the elongating spit while large dunes can develop on the updrift side.*

L485 "taller": (word choice) "high"

Variations on 'tall' and 'short' have been replaced by 'high' and 'low,' respectively. This change is throughout the manuscript.

L491-493: Meaning unclear.

The text has been revised to improve clarity:

*Long-term shoreline change rate is the cumulative result of waves moving sediment on a daily basis, while short-term variations in shoreline position caused by storms are feasible.*

L494 "variations in PAIS shoreline position": (missing) "identified here"

The text has been updated to reflect this suggestion.

L504 "likely to recover": (missing) "respond to and"

The text has been updated to reflect this suggestion.

L509 "it follows that spatial": (missing) "understanding"

The text has been updated to reflect this suggestion.

L515-517: What is the practical implication for current practice (e.g. Bruun and derivatives)?

The text was added to expand on the implications, as per this suggestion:

*Predicting coastal change without accounting for all factors affecting formation and evolution, such as directional dependencies due to framework geology, are more prone to uncertainty, which can have important managerial applications.*

L519-520: The potential is certainly there in areas where such data exist and where its implications are understood. Nourishment is just one of several management tools that stand to be assisted by such understanding.

> The text has been revised to be more broad in its application to coastal "engineering and management" projects:

> *Given that framework geology influences beach and dune morphology along the coast, the methods and results of this paper represent an opportunity for managers to improve coastal engineering and management projects, such as beach nourishment.*

L527-530: Somewhat gratuitous and not necessary.

> This sentence has been removed.

**Response to RC2**

In the discussion, the authors go on to state that their study area hasn't been significantly impacted by a storm since 1999. Some discussion of the extent to which this assumption and ultimately their results would hold if a storm did impact the island and how their conceptual model might apply outside of the Gulf of Mexico would help to frame the scope of the audience for this contribution.

> Some brief discussion has been added to describe how the results would be affected by hurricane impacts:

> *If hurricanes had impacted PAIS more frequently, it is likely that the alongshore variations in dune morphology, which were initially set up by the paleo-channels, would have been reinforced. This is because areas set up as low dunes would be preferentially overwashed while areas of high dunes would be more resistant and resilient during and following a storm. Therefore, the impact of a hurricane would highlight alongshore variations in dune morphology set up by the paleo-channels.*

P1, L14: What does this mean? Is there another way to describe this relationship? Since this is the primary distinction from previous work and it is used several times in the abstract, the meaning should be immediately clear and/or defined at first use.

> The text has been revised to improve clarity:

*This paper builds on previous research by demonstrating that paleo-channels in the framework geology can have a directional influence on alongshore beach and dune morphology.*

P1, L19: Without reading the paper, the reader may not have the context for what this means. Use another word to describe what you mean..."peaks?"

While 'peaks' and 'flares' capture similar patterns, the authors believe that 'flares' is a better fit because 'peaks' tends to

P1, L20 and L23: Why not just "paleo-channels" here? The current phrasing is redundant.

The text has been revised to reduce redundancy:

*Many 'flares' in the LRD plots correlate to relict infilled paleo-channels, indicating that the framework geology has a significant influence on the morphology of PAIS. Barrier island surface morphology LRD is strongest at large paleo-channels and decreases to the north. The spatial patterns in LRD surface morphometrics and framework geology variations demonstrate that the influence of paleo-channels can be asymmetric where the alongshore sediment transport gradient was unidirectional during island development.*

P2, L3 and L5: Rewrite these two sentences to eliminate redundancy.

Line 5 has been revised to reduce redundancy:

*Barrier island geomorphology can exhibit considerable variability alongshore, leading to varying responses to storm activity that ultimately determines the response of the island to sea level rise. Understanding the source of variability in beach and dune morphology can provide insight into how the barrier island is likely to change in response to future storms and sea level rise.*

P2, L7: What is meant by "patterns of vulnerability?"

The text has been revised to improve clarity:

*Storm waves interact with the variable morphology of the nearshore, beach, and dunes to determine how vulnerability varies along a barrier island.*

P2, L10-11: I think you mean "where VARIABILITY IN geologic structure can RESULT FROM variations in..., yes?

The text has been updated based on this reviewer comment:

*In this paper, the term "framework geology" is defined as any subsurface variation in geologic structure, where variability in geologic structure can result from variations in sediment type (i.e. sand vs. silt), differences in compaction, or significant changes in the subsurface organic content or mineralogy.*

P2, L19-20: There are many other models that assume alongshore uniformity beyond the probabilistic ones listed here. Consider citing a few of those as well. Also, Long et al., 2014 is not a modeling paper... did you mean to cite something else?

Additional citations have been added, and Long et al., 2014 has been removed.

P2, L28-30: Citation?

The text has been updated to include a citation for this statement:

*The nearshore bathymetry impacts sediment transport gradients along the island, which has implications for beach and dune response and recovery following a storm (Brenner et al., 2018).*

P2, L30-33: Try to summarize the contributions from Houser, 2012 more succinctly (e.g., in one sentence) so as not to lose the focus of the paragraph.

Text was condensed based on the reviewer comment:

*Houser (2012) demonstrated that variations in shoreline change patterns, beach width, and dune height corresponded to ridges and swales at Pensacola, FL. Shoreline position was more stable along the ridges, resulting in a wider beach which provided more sediment for onshore winds to create taller and more persistent dunes (Houser, 2012).*

P3, L3-5: I do not understand this sentence. Reword.

This sentence has been split to improve clarity based on the reviewer comment:

*This implies that variations in the framework geology, such as paleo-channels, do not influence long-term shoreline change, but, as noted, shoreline change is influenced by the alongshore variation in beach and dune morphology. The dissipative behavior of shoreline change does not negate the importance of framework geology.*

P3, L6-9: Why is vegetation being brought up here? How does this sentence help you set up the scientific problem you're testing?

The text has been updated to clarify that, while vegetation distribution does affect dune morphology, framework geology is the dominant driver of broad-scale dune morphology:

*While alongshore variation in dune morphology is also influenced by the distribution of vegetation in both space and time (Lazarus et al., 2011;Lazarus, 2016;Goldstein et al., 2017), the self-organized behavior of the dune morphology is ultimately set up by the framework geology (see Houser, 2012;Weymer et al., 2015b;Stallins and Parker, 2003).*

P3, L11-13: Regarding the stated purpose of this paper, how is this different than the purpose of your 2018 paper in Marine Geology, which states: "The purpose of this paper is to examine the influence of framework geology on beach and dune geomorphology at Padre Island National Seashore (PAIS), Texas, USA, where the framework geology is variable alongshore."? Use the introduction of this paper to make those distinctions explicit.

> The text has been updated to clarify that this paper is specifically focusing on the potential for asymmetries in beach-dune morphology that stem from paleo-channels in the framework geology:
>
> *The purpose of this paper is to test the hypothesis that relict infilled paleo-channels in the framework geology of a barrier island play a significant role and have an asymmetric influence on the alongshore variation in beach and dune morphology at a range of alongshore length scales.*

P5, L3-5: First mention of economic forecast models! Ok, so what is the purpose of this paper? It sounds like it is to determine how paleo-channels have influenced beach and dune evolution. But this isn't a process-based study, so how are you going to determine "the how?" It seems more likely that what you are trying to do is to adapt an economic forecast model to explore the spatial relationships (Are they lagged spatially? Is there a scale-dependence?) between beach and dune morphology and the presence/absence of subsurface paleochannels.

> The paper utilizes economic forecast models to explore the spatial relationships and (potential) lag between multiple variables in order to better understand how features such as paleo-channels have influenced coastal geomorphology.
>
> The text has been updated to clarify that identification of these spatial lags is the first step toward integrating them into future research models:
>
> *Identifying these spatial lags, their spatial scale(s), and their lag direction(s) is the first step toward integrating this information into morphodynamic prediction models.*

P5, L27-28: Here the distinction between the 2018 paper and this paper is clearly stated. But I would argue that Wernette et al., 2018 did more than "confirm the location of several paleo-channels." (That's actually what some of your co-authors papers, Fisk, and Anderson et al. did, no?) It also established a spatial connection between the presence of those paleochannels and beach and dune morphology by applying several signal analysis techniques in space. Be careful to distinguish this work from previous work, particularly Wernette et al., 2018.

The text has been updated slightly to make this distinction clearer:

*Previous research used EMI surveys to confirm the location of several paleo-channels and begin to quantify their influence on coastal geomorphology EMI surveys (Wernette et al., 2018;Weymer, 2016), while the current paper aims to determine the alongshore influence (direction and scale) of the paleo-channels.*

P7, L13-15: "Identifying the influence of very broad-scale influencing factors"? Rewrite this sentence to clarify what you mean. It's probably a good idea to change "framework geology" to "subsurface paleo-channels" or "paleo-channels" to help maintain the focus of THIS paper.

The text has been updated to specifically call out paleo-channels in the framework geology as a focus of the paper:

*In the case of coastal geomorphology, d parameter values may be particularly useful for identifying the influence of very broad-scale influencing factors, such as paleo-channels in the framework geology*

P7, L18-24: The distinction between this paper and previous work is clear in this section. Nicely stated. Should the focus of the paper be on the application of this technique to coastal geomorphology, specifically the relationship between paleo-channels and beach/dune morphology? The relationships identified in previous work could be used as justification for this "experiment." In fact, some evaluation of this technique over (or in place) of others previously used would be appropriate somewhere in the paper. In other words, could you just start with this analysis rather than those done in Wernette et al., 2018 or are those a prerequisite for ARFIMA modeling?

The text has been updated to clarify that the new approach presented can be used to identify the alongshore zone of influence:

*While wavelet decomposition can provide insight into relationships between two variables in the same location (Wernette et al. 2018), utilizing ARFIMA as a sliding window across multiple spatial scales can shed light on relationships that exhibit a lag in one or both directions. The advantage of this new approach is its application to examine alongshore influences of various natural and anthropogenic features (e.g. jetties, seawalls, groin fields, paleo-channels, and/or headlands) and identify their effective zone(s) of influence on coastal processes and geomorphology.*

The methodologies employed by the current paper and Wernette et al., 2018 are entirely independent and neither one relies on the other.

P7, L29-31: Complete this discussion with an explanation of how you arrived at the decision to use 250 unique computational windows; why not 200 or 300? How much do you have to increase or decrease the number of windows before you experience large increases in computing power or lose the ability to resolve the breakdown in spatial structure, respectively?

The challenge with describing computational performance is that it will vary significantly by machine. Describing, in detail, the specifics of the computational performance is not directly relevant to the purpose of the manuscript.

Evaluating each data series at 250 distinct spatial scales was selected because the available desktop computers were not able to handle 300 unique spatial scales. The authors aimed to maximize the number of unique alongshore spatial scales, which was approximately 250 scales.

The text has been slightly updated to highlight the intense computational requirements of analyzing a single data series:

*While the number of computational windows can be decreased, or increased, it is important to note that the ARFIMA modelling process is computationally intensive, requiring days to complete analysis of a single spatial data series on a high-performance desktop computer.*

P8, L33: Shouldn't the results section start with a description of plot 4a? Also, what is meant by "greatest LRD values?" Consider adding "as indicated by the dominance of red on the plot" or similar to keep the reader with you and reinforce the information from the previous section. Why doesn't apparent conductivity (Fig. 4a) show "the greatest LRD values?"

Based on another comment below the results have been restructured to discuss figure 4a and continue alphabetically. Text has also been updated to clarify what the authors mean by

*The shoreline change LRD plot exhibits the greatest LRD values (i.e. highest LRD values across all broad spatial scales) along the length of PAIS, as indicated by the dominance of red hues in figure 4b.*

P8, L34-P9, L3: Here and throughout the results section, it might be better to state your observations from the plots followed by your "interpretation" of the physical meaning. Here you might say, "Peaks in the shoreline change LRD plot are very narrow, which we interpret to mean that long-term shoreline change is dominantly dissipative...". Use the discussion to reference papers that support your interpretations or point out where they are not consistent with existing literature.

The text has been revised here and elsewhere throughout the Results, based on the suggestion provided:

*Peaks in the shoreline change LRD plot are very narrow, which we interpret to mean that the long-term shoreline change is dominantly dissipative with only minor undulations due to localized coastal processes, consistent with Lazarus et al. (2011) who demonstrated that broad-scale and long-term shoreline change is dissipative.*

P9, L11: Indicate on Fig. 4 where the island is divided into thirds (southern, central, and northern) so the reader can easily follow the text and identify the features to which you refer.

This is an approximation and has no clear division. As such, the authors have revised the text to clarify that this is a fuzzy boundary:

*Beach width LRD is more variable than shoreline change (Fig. 4c), with the least amount of variability concentrated in approximately the southern third of the island. These flares are likely present because…*

P9, L23-24: This is speculation. Move this to discussion where you can discuss your reasoning for this and support it with evidence from the literature.

This sentence has been removed from the Results. The Discussion already has a similar statement with supporting evidence.

P9, L33: How are dune crest elevation and dune height distinct from one another? Is there an advantage to using both rather than just one of these metrics?

Dune crest elevation is simply the elevation of the crestline, whereas dune height is the difference in elevation between the dune crest and dune toe. Dune crest elevation is a greater determining factor of whether a dune will be overwashed or inundated, whereas dune height is a better proxy for how developed a dune is. The advantage of using both is examining how developed the dune system is (through dune height) and its ability to resist overwash (dune crest elevation).

P10, L16 & P10, L23-24: Follow the first sentence of this paragraph with your interpretation of what it means (the last sentence of the paragraph) and then go in to the details of the broad- and fine-scale patterns that drives your interpretation. This structure could be applied to the entire results section, particularly the next paragraph (P10, L26-34).

The first sentence has been restructured to improve clarity as suggested by this comment:

*Island width exhibits the greatest alongshore variability in LRD of all island and framework geology morphometrics (Fig. 4g) and demonstrates that island width is dependent on broad- and fine-scale patterns of change.*

P11, L1-9: Either move this paragraph to the beginning of the results section to be consistent with the figure or change the figure to be consistent with the text. Also, the logic in this paragraph is a bit jumbled. Rewrite to follow the structure of observation/interpretation/reasoning.

The paragraph has been moved closer to the beginning of the Results section and slightly restructured to follow the suggested structure:

*Subsurface apparent conductivity exhibits substantial LRD along the entire length of PAIS (Fig. 4a). Patterns in the subsurface framework geology LRD plot demonstrate that the framework geology is self-similar at broader scales, and that this structure varies alongshore at finer alongshore length scales which correspond to the scale of the previously identified paleo-channels. The large LRD values at broad spatial scales (Fig. 4a) demonstrate that the paleo-topographic structure dominated by broad-scale coastal curvature over very broad spatial scales. Since the framework geology reflects the paleo-topography and the modern barrier island surface is dissipative at very broad scales, based on large LRD values at broad scales in the modern barrier island morphology, it follows that the framework geology is dissipative. The substantial LRD along much of the island supports previous work by Weymer (2016) and Weymer et al. (in press), which demonstrated that subsurface framework geology exhibits LRD at discrete locations and alongshore length scales.*

P11, L4-7 (highlighted in green): Are there patterns in Fig. 4a? If so, they are not apparent in the figure. Or is it the lack of any patterns (e.g., red everywhere) that you're referring to? Also, the part of the sentence that states "this structure varies very little alongshore and with scale" seems to contradict your statements elsewhere in this paper and in Wernette et al., 2018 that framework geology varies alongshore. Clarify. Finally, what do you mean by "the paleo-topographic structure trends toward a homogenous surface at broad scales?" So, if you shrunk the alongshore length scale the trend would be different?

> The text has been updated to improve clarity:

> *Patterns in the subsurface framework geology LRD plot demonstrate that the framework geology is self-similar at broader scales, and that this structure varies alongshore at finer alongshore length scales which correspond to the scale of the previously identified paleo-channels.*

> The text has also been updated to clarify that the "homogeneous surface" is actually in reference to the broad-scale coastal curvature:

> *The large LRD values at broad spatial scales (Fig. 4a) demonstrate that the paleo-topographic structure dominated by broad-scale coastal curvature over very broad spatial scales.*

P11, L12-25: This paragraph seems an odd fit here. It seems as if it's defending the choice of barrier island metrics used in the paper, which, if necessary at all, might be better suited for section 2.2. If that's not the point of this paragraph, it should be revised for clarity.

> This paragraph has been moved to section 2.2, as the authors agree it is more focused on describing the importance of the selected metrics.

P11, L30-31: The point about the depth of the paleo-channels is worth making earlier in the paper. If I understand what you're proposing correctly, the relationship that you're trying to tease out is the relationship between paleo-channels and barrier island morphology as the paleo-channels were infiling and (perhaps?) as the barrier island was forming. And that you are using the assumption that those dune heights/elevations initially established are what's driving the modern morphology now (see P11, L6-25), based on Houser, 2012; Weymer et al., 2015b; and Lazarus, 2016. This is quite the hypothesis, particularly since it would also assume that the barrier island had the same configuration as it does today (which is likely not the case). How can you better convince the reader that your hypothesis is supported by your analysis?

Text has been added to clarify that the authors are referring to the idea that larger paleo-channels (width and depth) affect beach-dune morphology along greater stretches of coastline than smaller paleo-channels. Furthermore, the zone of alongshore influence is because channel dimensions will dictate the accommodation space present and a paleo-channel's ability to serve as a sediment sink.

*In this way, larger paleo-channels (depth and width) will have a greater accommodation space and influence beach-dune morphology along a greater stretch of coast, while smaller paleo-channels have a more limited accommodation space and, therefore, influence a smaller stretch of adjacent coastal morphology.*

P12, L10-11: The use of "asymmetrically" here is confusing. I think what you mean is that paleo-channels and alongshore currents interact to drive asymmetries in their spatial relationship(s) to barrier island morphology, yes?

The text has been updated to clarify this confusion:

*This paper presents new information supporting the hypothesis that paleo-channels in the framework geology interact with alongshore currents to drive asymmetries in barrier island geomorphology and that the scale of influence is ultimately limited.*

P12, L18-24: The end of this paragraph is jumbled. Rewrite for clarity. It seems that ultimately this is a discussion of sediment supply as the channel was infiling and the island/beach/dunes were forming. I think what you're trying to get at is that oblique channels would have taken more sediment to fill relative to shore-normal paleo-channels, thereby leaving less sediment available for building beaches and dunes?

The text has been revised to improve clarity:

*An oblique channel would have required more sediment and take longer to fill than a shore-normal channel. Subsequently, a wide beach and dunes would begin to form in the shore-normal paleo-channel before the oblique paleo-channel. For an oblique paleo-channel the volume of sediment required to fill the channel from*

*alongshore sediment transport and fluvial deposition from the mainland would likely have been insufficient to build a wide beach to supply sediment for significant dune growth.*

P12, L26- P13, L12: This section would benefit from a schematic of the processes you describe. Where is sea level when these processes are ongoing? Is this consistent with published theories of coastal evolution for the area (a la Weise and White, 1980)?

> Information about sea-level has been added to this description. A schematic of the system formation and evolution has been added as figure 7.

P13, L3-5: Citation for the baymouth-bar analogy?

> This sentence has been revised to exclude reference to baymouth bars:

> *The closest modern analogy to this alongshore sedimentation process is the formation and evolution of an alongshore spit eventually completely crossing the outflowing river channel, where the river is eventually cut off by the elongating spit. In this case, sediment is supplied to the updrift beach and provides a sediment source for dunes to form.*

Figure 2. Reference to "oblique gravel ridges" in caption of panel c is incorrect. Gravel was present in the troughs between sandy shore-oblique bars. Suggest a citation for Browder and McNinch, 2006 here instead.

> The figure caption has been updated to "oblique sandbars," as the features are originally referred to by McNinch (2004).

Figure 3. This figure appeared to be ~1.5 inches high by 2 inches wide in the submitted manuscript. Consider making this figure much larger so the reader can follow your interpretation of the results, particularly with regard to asymmetry. For example, is your interpretation of asymmetry due to what appears to be a "tail" that trails off to the left at the top of the peak at "B" or due to the concentration of darker blue toward the left of the bottom of the right peak under "B?"

> This is the correct interpretation, as also described in the text. The figure has been increased in size for legibility purposes.

Figure 4. It's very difficult to see the labels on the x and y axes in all of these plots. It could be separated into two figures, one with AC, Shoreline Change, 2km Bathy, and 4km Bathy and a second with all of the island metrics (BW, DTE, DCE, DH, and IW).

> The authors would prefer to keep the figures as one because it facilitates comparison of one LRD plot against another LRD plot at a given location alongshore. Splitting the figure into 2 separate figures would make this comparison substantially more difficult.

Figure 5. I don't understand the justification for the coloration of the northern half of these panels with the SRD polygon. What are you looking at to justify the northward scale decrease? In panel B, there's a peak about an inch in from the left that's taller than the peaks south of it which would seem to contradict the interpretation as I understand it.

The colors are used to highlight a general trend in the transition from LRD to SRD (as marked by the blue-red boundary). The gradual ramp up from north to south is based on a generalization of the flares present in the LRD plots. Although there are other peaks in the data, they are highly localized and, when generalizing the plots, they do not fit into any larger patterns. This is similar to filtering the data for low-frequency patterns.

[revised manuscript text omitted]

---

## Author Response (AR2)

I would like to see a complete reply to Ref.2 comments (e.g. second and third paragraphs), in particular I would like a clear answer to the question of what are the differences--in methods, results and interpretations--between this contribution and Wernette et al., 2018.

**Author Response:**
**Summary:** The current manuscript is distinct from Wernette et al., 2018 in methodology, results, and interpretations. While the following information was detailed in the previous response to reviewer comments, the current response to the editor feedback is meant to better articulate the several important distinctions between the current manuscript and Wernette et al., 2018. Since all of the following information and edits were included in the previous rounds of edits in response to Reviewer 1 and 2 comments, the manuscript remains unchanged from the previously revised version.

Using ARFIMA modelling, not wavelet decomposition or bicoherence, the current manuscript demonstrates that paleo-channels in the framework geology affected alongshore patterns in beach and dune morphology within distinct zones of influence, while Wernette et al., 2018 was focused on confirming the location of paleo-channels and their impact on beach-dune morphology only within a given channel. The previous paper did not explore what the potential alongshore zone of influence was for a given paleo-channel because the methodology was incapable of such analyses. Specifically, the current manuscript is the first paper to identify the alongshore distances that paleo-channels affected beach-dune morphology, as noted in the revised text:

> (Lines 86-87) *"The purpose of this paper is to test the hypothesis that relict infilled paleo-channels in the framework geology of a barrier island play a significant role and have an asymmetric influence on the alongshore variation in beach and dune morphology at a range of alongshore length scales."*

As described in the previously updated text and responses to reviewer comments, the Abstract was revised to articulate the differences between the current manuscript and Wernette et al., 2018:

> (Lines 6-7) *"This paper builds on previous research by demonstrating that paleo-channels in the irregular framework geology can have a directional influence on alongshore beach and dune morphology."*

> (Lines 18-20) *"The spatial patterns in LRD surface morphometrics and framework geology variations demonstrate that the influence of paleo-channels can be asymmetric (i.e. affecting beach-dune morphology preferentially in one direction alongshore) where the alongshore sediment transport gradient was unidirectional during island development."*

**Differences in Methodologies:** Previous work was focused on confirming the location of paleo-channels in the framework geology and examining how these infilled channels affected the modern barrier island geomorphology *within the channels themselves*. Neither wavelet decomposition nor

bicoherence analyses, as used in Wernette et al., 2018, are capable of being used to determine the alongshore zone of influence for a given feature. Wavelet decomposition only provides information on relationships between co-located variables, and bicoherence only examines frequency relationships within one or more data series. ARFIMA models, as used in the current manuscript, can be used to identify how far alongshore a given paleo-channel influenced barrier island geomorphology. ARFIMA is completely different than wavelet decomposition and bicoherence techniques, does not rely on any of Wernette et al., 2018 methods, and is capable of distinguishing between fine- and broad- scale coastal processes and geomorphology.

The previously updated manuscript and response to reviewer comments included differentiation between the methodologies of Wernette et al., 2018 and the current manuscript:

> (Lines 158-160) *"The current paper expands on previous research by adapting economic forecast models to determine how paleo-channels in the framework geology have influenced beach and dune evolution and whether this influence is directional and scale-dependent."*

> (Lines 208-211) *"Previous research used EMI surveys to confirm the location of several paleo-channels and begin to quantify their influence on coastal geomorphology EMI surveys (Wernette et al., 2018; Weymer, 2016), while the current paper aims to determine the alongshore influence (direction and scale) of the paleo-channels."*

> (Lines 287-293) *"While wavelet decomposition can provide insight into relationships between two variables in the same location (Wernette et al., 2018), utilizing ARFIMA as a sliding window across multiple spatial scales can shed light on relationships that exhibit a lag in one or both directions. The advantage of this new approach is its application to examine alongshore influences of various natural and anthropogenic features (e.g. jetties, seawalls, groin fields, paleo-channels, and/or headlands) and identify their effective zone(s) of influence on coastal processes and geomorphology."*

**Differences in Results:** Although the ARFIMA plots appear similar in structure to wavelet plots, the methodology used to derive the values and values themselves are completely different than wavelet decomposition and bicoherence analysis. The current manuscript isolates the $d$ parameter in ARFIMA models fit to barrier island geomorphology data series as a way of identifying the alongshore zone of influence for paleo-channels in the framework geology. The simplest approach to interpreting variations and patterns in the $d$ parameter values is by visualizing it similar to a wavelet decomposition plot, with scale on the y-axis and alongshore location on the x-axis.

**Differences in Interpretations:** As described in the updated text and in response to the previous reviewer comments, the previously updated manuscript describes the differences between the current paper and previous work at the following locations:

(Lines 158-160) *"The current paper expands on previous research by adapting economic forecast models to determine how paleo-channels in the framework geology have influenced beach and dune evolution and whether this influence is directional and scale-dependent. Identifying these spatial lags, their spatial scale(s), and their lag direction(s) is the first step toward integrating this information into morphodynamic prediction models."*

(Lines 287-293) *"While wavelet decomposition can provide insight into relationships between two variables in the same location (Wernette et al., 2018), utilizing ARFIMA as a sliding window across multiple spatial scales can shed light on relationships that exhibit a lag in one or both directions. The advantage of this new approach is its application to examine alongshore influences of various natural and anthropogenic features (e.g. jetties, seawalls, groin fields, paleo-channels, and/or headlands) and identify their effective zone(s) of influence on coastal processes and geomorphology."*

(Lines 346-348) *"Patterns in the subsurface framework geology LRD plot demonstrate that the framework geology is self-similar at broader scales, and that this structure varies alongshore at finer alongshore length scales which correspond to the scale of the previously identified paleo-channels."*

(Lines 465-467) *"This paper presents new information supporting the hypothesis that paleo-channels in the framework geology interact with alongshore currents to drive asymmetries in barrier island geomorphology and that the scale of influence is ultimately limited."*